# Functional and structural consequences of epithelial cell invasion by *Bordetella pertussis* adenylate cyclase toxin

Christelle Angely[1,2,3], Daniel Ladant[4], Emmanuelle Planus[5], Bruno Louis[1,2,3], Marcel Filoche[1,2,3,6], Alexandre Chenal[4], Daniel Isabey[1,2,3]*

**1** Equipe 13, Biomécanique & Appareil Respiratoire, Inserm U955, Créteil, France, **2** UMR 955, UPEC, Université Paris-Est, Créteil, France, **3** ERL 7000, CNRS, Créteil, France, **4** Unité de Biochimie des Interactions Macromoléculaires (CNRS UMR 3528), Département de Biologie Structurale et Chimie, Institut Pasteur, Paris, France, **5** Institut pour l'Avancée des Biosciences (IAB), Centre de Recherche UGA/ Inserm U1209 / CNRS UMR 5309, La Tronche, France, **6** Laboratoire de Physique de la Matière Condensée, Ecole Polytechnique, CNRS, IP Paris, Palaiseau, France

* daniel.isabey@inserm.fr

**Data Availability Statement:** All relevant data are within the paper.

**Funding:** DI: grant number DBS20140930771 Fondation pour la Recherche Médicale http://www.

## Abstract

*Bordetella pertussis*, the causative agent of whopping cough, produces an adenylate cyclase toxin (CyaA) that plays a key role in the host colonization by targeting innate immune cells which express CD11b/CD18, the cellular receptor of CyaA. CyaA is also able to invade non-phagocytic cells, via a unique entry pathway consisting in a direct translocation of its catalytic domain across the cytoplasmic membrane of the cells. Within the cells, CyaA is activated by calmodulin to produce high levels of cyclic adenosine monophosphate (cAMP) and alter cellular physiology. In this study, we explored the effects of CyaA toxin on the cellular and molecular structure remodeling of A549 alveolar epithelial cells. Using classical imaging techniques, biochemical and functional tests, as well as advanced cell mechanics method, we quantify the structural and functional consequences of the massive increase of intracellular cyclic AMP induced by the toxin: cell shape rounding associated to adhesion weakening process, actin structure remodeling for the cortical and dense components, increase in cytoskeleton stiffness, and inhibition of migration and repair. We also show that, at low concentrations (0.5 nM), CyaA could significantly impair the migration and wound healing capacities of the intoxicated alveolar epithelial cells. As such concentrations might be reached locally during *B. pertussis* infection, our results suggest that the CyaA, beyond its major role in disabling innate immune cells, might also contribute to the local alteration of the epithelial barrier of the respiratory tract, a hallmark of *pertussis*.

## Introduction

The adenylate cyclase (CyaA) is a major toxin secreted by *Bordetella pertussis*, the causative agent of whopping cough. This toxin plays a key role in the early stage of colonization of the respiratory tract by *Bordetella pertussis*. CyaA is able to invade eukaryotic cells in which it

frm.org The funder had no role in study design, data collection and analysis, decision to publish, or preparation of the manuscript.

**Competing interests:** The authors have declared that no competing interests exist.

translocates its catalytic domain which is activated by endogenous calmodulin to catalyze a massive production of cyclic AMP (cAMP), resulting in profound alterations of cellular physiology [1–3]. The cytotoxic effects of CyaA are mainly directed towards neutrophils and macrophages, as well as other innate immune cells that express the CD11b/CD18 integrin, which is the main cellular receptor of CyaA [4]. CyaA inhibits the phagocytic function of neutrophils and macrophages by impairing chemotaxis and oxidative response, and eventually triggers cell apoptosis and necrosis [5–9]. Yet, CyaA is also able to invade a wide variety of non-phagocytic cells [1, 2, 8]. A similar unique mechanism of entry in the cytosol is likely used by CyaA in both immune and non-immune cells. It consists in a direct translocation of the catalytic domain across the cytoplasmic membrane of the cells [10] which is confirmed by the rapidity of the intoxication (or internalization) process (basically a few seconds) with however differences depending on cell type [10–12]. Nevertheless, the molecular mechanisms by which CyaA penetrates the target cells are still largely elusive and the physiological consequences of the toxin activity remain to be precisely determined especially in non immune cells [1]. In particular, several studies have recently suggested that CyaA might also act at the site of infection on the epithelial cells of the respiratory tract. First, Eby *et al.* [13] have quantified the amounts of CyaA that is produced locally during infection of the respiratory tract and they found that, at the bacterium-target cell interface, the concentration of CyaA may exceed 100ng/ml (about 0.6nM of toxin). Other studies have suggested that CyaA at these concentrations may affect the epithelial cells and eventually contribute to the alteration of the epithelial barrier which is primarily affected by the cytopathic action of the tracheal cytotoxin, TCT, released by *B. pertussis*. Ohnishi *et al.* (2008) first showed that CyaA induces morphological changes on cultured rat alveolar epithelial cells [14]. Eby *et al.* reported that polarized T84 cell monolayers as well as human airway epithelial cultures could respond to nanomolar concentrations of CyaA when it was applied to the basolateral membranes [15]. We then showed that CyaA can invade A549 alveolar epithelial cells and trigger a significant remodeling of their molecular adhesion systems [16] and, more recently, Hasan *et al.* (2018) analyzed the impact of CyaA on the functional integrity of human bronchial epithelial cells cultured at the air-liquid interface [17].

Cytoskeletal (CSK) rearrangement may be caused by different signal events including modifications in the intracellular $Ca^{2+}$ and cAMP-mediated signaling which have been involved in perturbations of the actin CSK homeostasis in different cells [18, 19]. These two factors are indeed deeply modified and markedly increased after CyaA invasion suggesting that the CyaA toxin could trigger significant CSK remodeling of the host cells [8, 9, 20, 21].

In the present study, we further characterized the CyaA-induced structural remodeling and functional alterations of the CSK and mechanical properties of A549 alveolar epithelial cells. We show that CyaA intoxication reduces the migration and wound healing capacities of these cells when they are exposed to toxin doses mimicking those reached in vivo during *B. pertussis* infection. Our present results therefore suggest that the CyaA toxin may contribute to the local disruption of the integrity of the airway epithelium.

## Materials and methods

### Cellular model of intoxication

**Culture of Alveolar Epithelial Cell lines (AECs).** Experiments were carried out on A549 cells which are an alveolar epithelial cell line (AECs) classically used for cell respiratory physiology studies. Briefly, this line, which originates from a pulmonary epithelium adenocarcinoma taken from patient, is obtained from the National Cancer Institute's lineage library (ref: ATCC Collection No. CCL-185). A549-type epithelial cells have been used in the laboratory

for many years [22, 23] as they express a phenotype like certain pulmonary alveolar epithelial cells, i.e., the type II pneumocytes [24].

AECs offer many advantages for studying in vitro the pathophysiological response of pulmonary cells [25]. They form adherent and tight junctions when grown to confluence and express a wide variety of cytokines, growth factor and receptors and notably several transmembrane receptors of the integrin type [26]. These integrin receptors bind the synthetic peptide containing the RGD sequence present in many extracellular matrix components. The peptide RGD is classically used for integrin-specific cell-binding as done in the present study and in many previous studies [27, 28]. To maintain integrin expression at a sufficiently high level [29], the passage number was maintained in the low range ($\approx$12th–16th).

The cells are cultured in plastic flasks treated for cell adhesion with a filter cap (25 or 75 cm$^2$, Techno Plastic Products AG, Switzerland). The culture medium consists of DMEM (Gibco Life Technologies), 10% fetal calf serum or FCS (Sigma-Aldrich, St. Louis, MO, USA) as well as 1% antibiotics (penicillin and streptomycin). The FCS is the most complex component because it contains growth factors, hormones, elements of the extracellular matrix, e.g., fibronectin and vitronectin, and all other element contained in the blood, except the figured elements, i.e., the coagulation factors and the complement.

The cultures are incubated at 37°C in a controlled atmosphere (5% $CO_2$ and 95% humidity). The cells are adherent to the support and must therefore be peeled off using trypsin-EDTA 0.05% (Sigma-Aldrich, St. Louis, MO, USA) and then subcultured with a split ratio of 1/10. After centrifugation at 200g, the cell pellet is re-suspended in DMEM-10% FCS medium and a part is transferred to another flask. To keep the line and have a stock, the unused cells are frozen. For experiments, we used a density of $7.10^5$ cells and seeded on petri-dish (TPP $\emptyset$34mm) coated with fibronectin at 10ng/mm$^2$.

**Production and purification of CyaA and its inactive variant CyaAE5.** CyaA and the enzymatically inactive variant CyaAE5 (resulting from a Leucine and Glutamine (LQ) dipeptide insertion between D188 and I189 in the catalytic core of the enzyme [30]) were expressed in *Escherichia coli* and purified to homogeneity by previously established procedures [4, 31]. Succinctly, the inclusion bodies were solubilized overnight at 4°C in 20mM Hepes, 8M urea, pH 7.4. The soluble urea fraction was recovered by centrifugation, supplemented with 0.14M NaCl and then loaded on a Q-Sepharose fast flow resin equilibrated with 20mM Hepes, 140 NaCl, 8M urea, pH 7.4. Contaminants were eliminated by an extensive wash in the same buffer; the CyaA protein is then eluted using a NaCl gradient (CyaA elution occurs around 500mM NaCl). After dilution of the eluate in 20mM Hepes, 8M urea to reach 100mM NaCl, the CyaA batch is loaded onto a second Q-sepharose Hi Performance column. Washing and elution are operated in the same conditions as the first Q. This step allows to obtain a concentrated CyaA protein which is then diluted 5 times with 20mM Hepes, 1M NaCl, pH 7.4 and loads onto a 70ml phenyl-sepharose column and washes with 20mM Hepes, 1M NaCl, with Hepes 20mM again and then with 50% isopropanol. After an extensive wash, the toxin is eluated with 20mM Hepes and 8M urea. The eluate is then applied onto a sephacryl 500 (GE healthcare, HIPREP 26/60) equilibrated in 20mM Hepes and 8M urea. CyaA batches are pooled and concentrated by ultrafiltration and stored at -20°C in 20mM Hepes and 8M urea. CyaA toxin concentration is determined by UV-spectrophotometry using a molecular extinction coefficient Em280 = 143590 M$^{-1}$cm$^{-1}$ computed from the CyaA sequence on the Prot-Param server (http://web.expasy.org/protparam/http://web.expasy.org/protparam/). The purity of CyaA batches is higher than 90% as judged by SDS PAGE analysis and contained less than 1 EU of LPS/µg of protein as determined by a standard LAL assay (Lonza). Finally, CyaA is refolded into a urea-free, monomeric and functional holo-state [31, 32]. The refolding efficiency is around 40±5% (population monomer / total population of proteins). The biological

functions, i.e., hemolysis, translocation of Adenylate Cyclase Domain (ACD) and cAMP production, are routinely assayed as described in [31] and [33]. The aliquots of monomeric species of CyaA are stored at -20°C in 20mM Hepes, 150mM NaCl, 2mM $CaCl_2$, pH 7.4.

**Handling of CyaA and CyaAE5 toxins.** For our experiments, CyaA and CyaAE5 recombinant proteins are diluted into a mix solution containing DMEM-10% Fetal Bovine Serum (FBS), buffer and CaCl2. CyaA mixture or CyaAE5 mixture is directly added on cells for 30 or 60 min. We used a range of concentration of CyaA or CaAE5 toxin from 0.5 to 10nM.

## Biological tests

**cAMP assays.** A549 cells are seeded at $3.10^5$ cells/well in 96 well plates. After 24 hours, cells are exposed for 60 min to three different concentrations (0.5, 5, and 10nM) of either the CyaA toxin or the enzymatically inactive CyaAE5 protein. The CyaA mix was removed and cells were washed. Cells were then recovered and lysed using 0.1M HCl in order to collect the intracellular content. The concentration of cAMP in the lysates was then measured by using a competitive ELISA kit for cAMP (Invitrogen, ref. EMSCAMPL) following the recommendations of the manufacturer.

**Cell viability evaluation by MTT and trypan blue tests.** MTT (3- [4,5-dimethyl-2-thiazolyl] -2,5-diphenyltetrazolium bromide, Sigma M5655) is a tetrazolium salt giving a yellow solution when diluted in the medium. It is converted into insoluble violet formazan in the culture medium after cleavage of the tetrazolium ring by the active mitochondrial dehydrogenases of living cells only (dead cells are not detected by this assay). 2500 cells are seeded per well in a 96-well plate. After 24 hours of incubation, the cells are synchronized with DMEM at 0% FCS without phenol red. After 72 hours of incubation, different concentrations of CyaA are added for 60 minutes. Then the CyaA mix was removed and cells were washed. Finally, 50 μL of MTT solution at 2 μg/ml are then added to each well. After 4 hours of incubation, the medium is removed by inversion. 200 μL of pure DMSO were then added to each well. Finally, the 96-well plate is read from the ELISA reader at a wavelength of 550nm. Results are expressed in terms of optical density OD (or absorbance).

The Trypan blue is a vital stain used to selectively color dead cells which is used complementary to the MTT test. The colorant penetrates and stays only in dead cells which are not able to eject it. This mechanism requires a cellular energy that is provided by mitochondria, thus enabling to eject the colorant but only in living cells. Practically, the Trypan blue (Sigma-Aldrich) is added to cell suspension which was previously exposed or not to CyaA toxin (0.5, 5 and 10nM). Then, specifically for the cell counting procedure, we discriminated live cells (no color) and dead cells (in blue) by counting with a hematocytometer Malassez to obtain a viability percentage: (live cell number/total cell number) ×100.

**Actin cytoskeleton (CSK) and focal adhesion staining.** The A549 cells are seeded at a density of 75,000 cells per coverslip (∅12mm). After 24h incubation at 37°C and 5% $CO_2$, the cells reach 60% confluence and are incubated for 60 min at 37°C in complete medium (control conditions) or complete medium supplemented with CyaA toxin (0.5, 5 or 10nM final concentrations). In some experiments, in control and after 60 min of CyaA incubation, cells are exposed 30 min to $Mn^{2+}$ (0.5mM final concentration). The cells are washed with PBS and then fixed for 10 min in 4% paraformaldehyde, permeabilized by addition of 0.3% Triton in PBS. Non-specific sites are blocked by incubation of PBS-1% BSA during 30 min. Then, cells are incubated for 1 hr at room temperature with anti-phosphotyrosine primary antibody PY99 (sc-7020, Santa Cruz Biotechnology) diluted at 1/150 in PBS-1% BSA. The PY99 antibody is then detected with an Alexa-488 labeled secondary antibody (Life Technology ref. A-21042 diluted 1/1000 in PBS) to reveal focal adhesion points (in green). After rinsing with PBS-

0.02% Triton, the cells are incubated in a dilute solution of phalloidin tetramethylrhodamine bisothiocynate (Sigma, ref. P1951 diluted at 1/2000 in PBS) in order to display actin fibers (in red).

Coverslips were mounted on a slide with the cell side down in Prolong (LifeTechnology, ref. P36974). Staining structures are observed, and images are acquired on an Axio Imager confocal microscope (Zeiss) at 63× magnification. To analyze images of the cells selected by random fields, the mean level of fluorescence of F-actin is determined in each cell by automatic quantification by ImageJ software relatively to the fluorescent background level of each image. The effective mean fluorescence of cells is obtained after subtraction of background fluorescence. In order to distinguish the cortical and the dense actin filaments, two different fluorescence thresholds on ImageJ software were used, i.e., 152 for the cortical actin and 534 for the dense actin, to permit the quantification of actin fluorescence at two detected levels. In the same way, different thresholds were used to distinguish different size areas of focal adhesion points (< or > to 1 μm$^2$). Finally, on the same coverslips, we used the LSM5 Pascal software to make stacks of images (every 0.5 μm) in the vertical direction 'z' and to carry out a 3D reconstruction of the shape of the cell in order to measure the cell height in each condition.

## Functional assessments

**Measure of the CSK stiffness by Magnetic Twisting Cytometry (MTC).**  A549 cell mechanical properties were measured by a bead micromanipulation system through CSK specific probing called MTC which has been extensively described in the literature [27, 28]. Probes are ferromagnetic microbeads functionalized with an integrin specific ligand, e.g., fibronectin. When an external magnetic field is exerted, these beads apply a torque to the living cell structure through CSK specific-ligand binding. The relationship between the measured bead rotation angle (angular strain averaged over the cell culture during cellular loading), and the applied torque provides CSK-specific measurements of mechanical (rheological) properties of the cell. This technique invented by Wang *et al.* in 1993 [28] has been recently upgraded by Isabey and co-workers to achieve a multiscale quantification of cellular and molecular parameters [27]. Each MTC measurement is performed twice on a wide number of cells ($7 \times 10^4$). The specific attachment of microbeads to the actin CSK is assessed in control conditions by treating cells with the actin-depolymerizing drug Cytochalasin D (CytoD) at a concentration of 10 μM. Advantageously, the large number of beads and their uniform distribution throughout the cell culture guarantee an instantaneous homogenized cell response, representative of the dominant structural behavior. The key parameter obtained is the elasticity modulus or cytoskeleton stiffness (*E* in Pa), which has been measured for the different concentrations of the CyaA toxin. *E* quantifies the mechanical stiffness of CSK structure and reflects also the intracellular tension or prestress [34].

It should be underlined that, based on the principle of the MTC method presently used, only living cells contribute to the MTC signal. Indeed, cell stiffness vanishes when cells are dead, (e.g., acto-myosin motors are not active anymore and internal tension drops to zero). Thereby ferromagnetic microbeads—if still attached to cell surface—would turn instantaneously by 90° and thus would not contribute to the MTC signal. Similarly, detached beads floating in the culture medium, would turn freely and thus would not contribute to the signal either [27].

**Measure of cell migration and wound healing.**  A549 cells were seeded at a density of $10^5$ cells/chamber in a 12-well simple chamber. There were incubated during 24 hours in complete medium (DMEM, 1% penicillin, 5% SVF) to form a confluent monolayer. Cells were exposed during 1 hr to the different concentrations of CyaA toxin or to complete medium (control

conditions). The cell monolayer was scratched with a pipette tip of 5 μm-distal diameter to create a wound and rinsed 3 times with medium in order to remove the entire detached cell population. Transmission images of the wound repair of the cell monolayer are taken every 3 hrs with a black and white CDD camera (Kappa DX20-H) mounted on a fluorescent-transmission microscope (Leitz Labovert FS). Cell migration was determined by the aptitude of cells to close the wound. The quantification of the repair of the monolayer is made by measuring the wound area using ImageJ (W.S. Rasband; NIH, Bethesda). The evolution of the repaired area is quantified by the following formula:

$$\text{Repaired area}(\%) = \frac{(\text{initial area} - \text{current area})}{(\text{initial area})} \times 100 \qquad (1)$$

The wound repair is followed until complete closure (repaired area = 100%).

## Results

We previously showed [16] that A549 alveolar epithelial cells are sensitive to CyaA that triggers a cAMP-dependent remodeling of their molecular adhesion system, resulting in a significant weakening in their initial adhesion processes. Here we further examined the effects of CyaA on the reorganization of the actin cytoskeleton and evolution of focal adhesions. A549 cells were exposed for 1 hr to various concentrations of CyaA (ranging form 0.5 to 10nM) and then the F-actin structures were visualized by phalloidin staining (in red) and focal adhesions were visualized with anti-phosphotyrosin PY99 antibodies (in green) (Fig 1).

In cells incubated with CyaA, a marked remodeling of actin structure and focal adhesion reorganization is observed. This effect is characterized by a progressive concentration-dependent disappearance of stress fibers associated with a significant loss of number of focal adhesion points. Noteworthy, the effect of CyaA intoxication on both actin structure and focal adhesion could be partially counteracted by addition of manganese (0.5mM $Mn^{2+}$) during the last 30 min of incubation with CyaA (Fig 1). Control cells exposed to $Mn^{2+}$ (Fig 1D) shows a strong reinforcement of both F-actin structure and most particularly focal adhesion points, as expected given the activation of integrins forced by $Mn^{2+}$ [35, 36]. It appears that CyaA-intoxicated A549 cells still respond to manganese exposure but to a lesser extent than the non-intoxicated (control) cells. Thus, $Mn^{2+}$ treatment enables partial reversibility of the intoxication effects induced by CyaA. Due to its high affinity for integrins [35, 36], $Mn^{2+}$ enables to reactivate adhesion and cell spreading via the reinforcement via the integrin activation and clustering at the cell membrane of intoxicated A549 cells. The $Mn^{2+}$ effects suggest that CyaA toxin affects preferentially both the actin CSK and the integrin-dependent adhesion system.

Quantifications of global fluorescent F-actin at the basal face of spread A549 after 60 min of CyaA exposure (Fig 2) reveal a significant decrease in F-actin intensity, in a toxin concentration-dependent manner. This confirms the microscopic observations shown above. Quantifications of F-actin structures at low density F-actin (cortical or sub-membranous F-actin) and high-density F-actin (stress fibers) reveal that cortical actin sub-structures are predominantly and significantly affected by CyaA exposure while the dense actin sub-structures are only slightly affected (Fig 2). The decrease in total F-actin intensity in CyaA-intoxicated cells is also observed after $Mn^{2+}$ exposure although the proportion of cortical versus dense F-actin is modified in favor of the dense F-actin with manganese (compare Figs 2 and 3).

Quantification of the number of focal adhesion (FA) points after 60 min of CyaA exposure is given for all FA sizes as well as for the small ($\leq 1 \ \mu m^2$) and large ($> 1 \ \mu m^2$) FA sizes (Fig 4). The total number of adhesion sites in cells exposed to CyaA markedly decreases as the toxin concentration increases. For all conditions studied, the wide majority of adhesion sites is

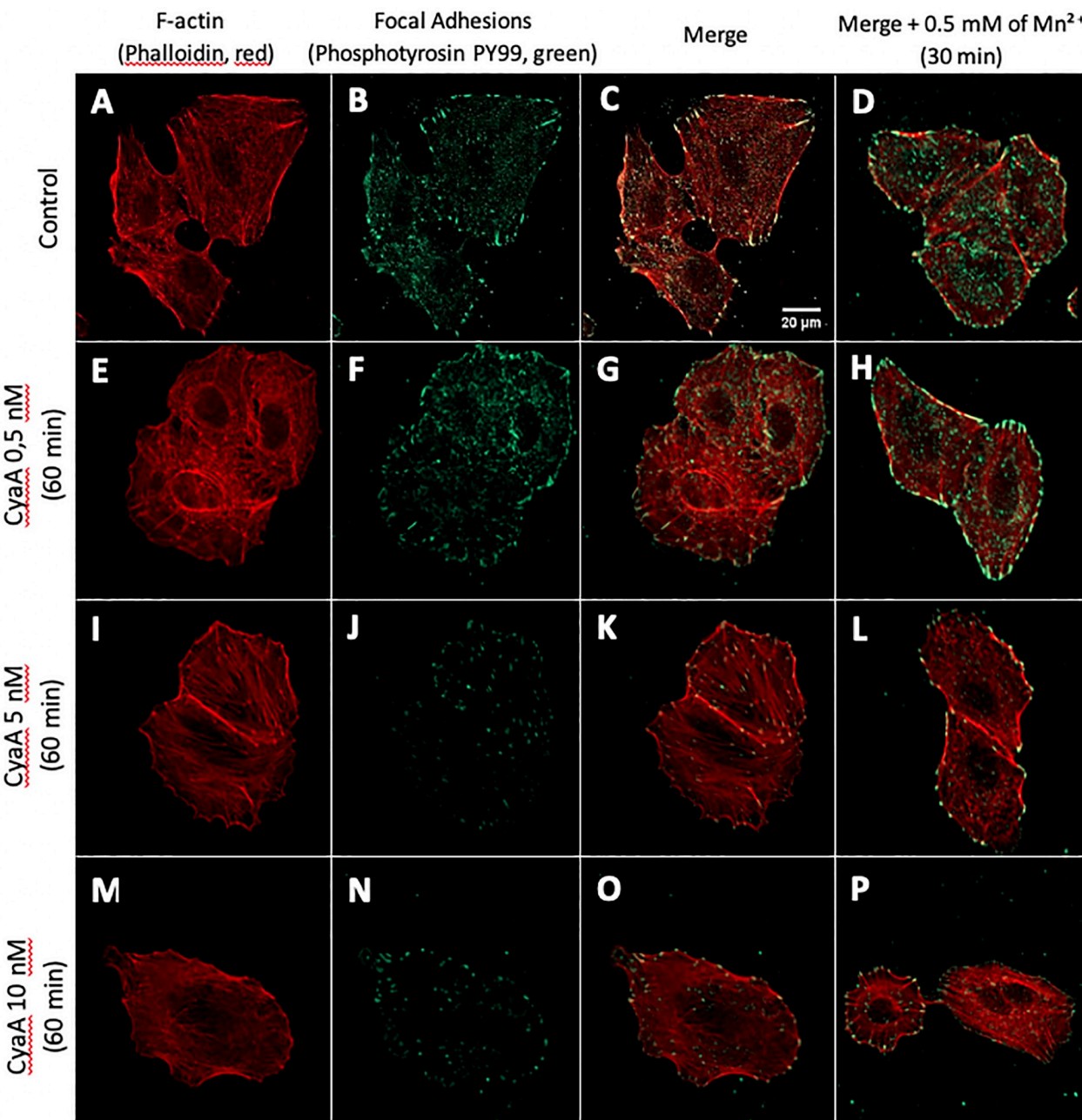

**Fig 1. Cell imaging of F-actin structure and focal adhesion points.** Co-staining of F-actin (with phalloidin, in red) and of focal adhesions (with anti-phosphotyrosin PY99 antibody, in green) in fixed A549 cells (control conditions) after 1 hr of exposure to the indicated concentrations of CyaA toxin. (A, E, I, M) F-actin staining, (B, F, J, N) focal adhesion staining, (C, G, K, O) merge images. Right panels (D, H, L, P) show merged images of F-actin staining and focal adhesion staining for cells incubated with the indicated concentrations of CyaA for 60 min followed by a 30 min exposure to $Mn^{2+}$ at a concentration of 0.5mM. (A-D) Control conditions (no CyaA), (E-H) cell exposure to 0.5nM of CyaA, (I-L) cell exposure to 5nM of CyaA, (M-P) cell exposure to 10nM of CyaA. Images were obtained by confocal microscopy with ×63 magnification. S1 and S2 Figs further document the cell viability and intracellular cAMP accumulation in the A549 cells exposed to CyaA.

constituted by small adhesion sites, the number of which is significantly reduced as the CyaA concentration increases. By contrast, the number of large adhesion sites is always significantly smaller than the total number of FA (statistics not presented) but, compared to small FA, the decay in the number of large FA (as the CyaA concentration increases) is only slightly

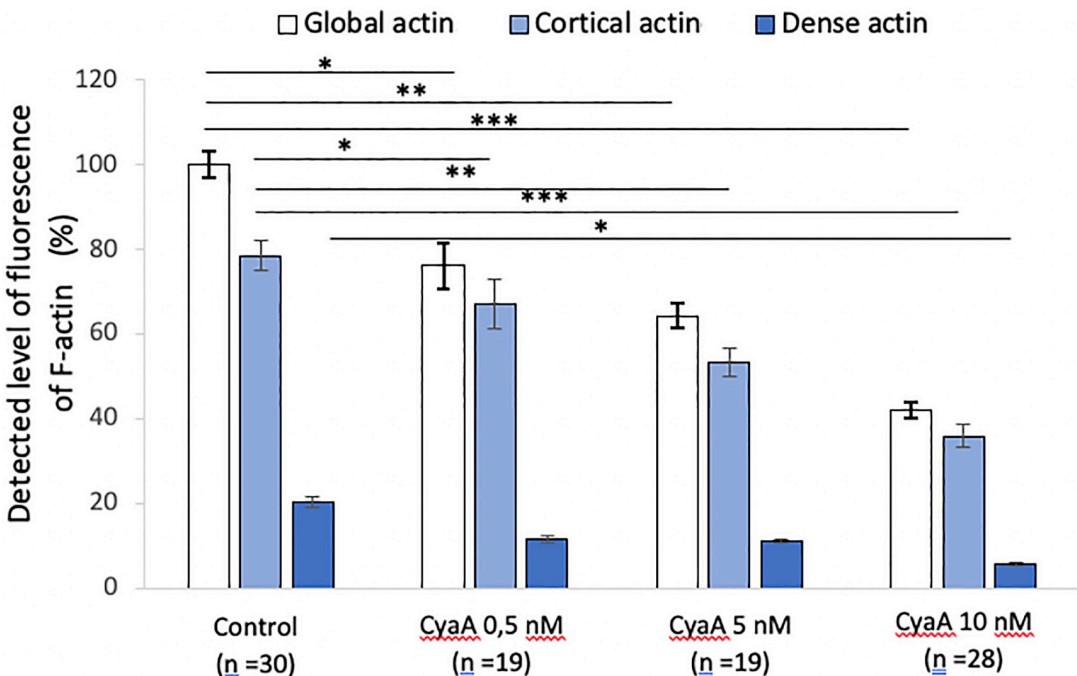

**Fig 2. Quantification of fluorescence levels of global, cortical and dense F-actin in A549 after 60 min of CyaA exposure.** The bar graph provides a quantification of F-actin fluorescence in A549 cells before (control conditions) and after 60 min of exposure to the indicated concentrations of CyaA. The specific distribution of the cortical (submembranous) and dense (stress fibers) actin structures were analyzed by using intermediate thresholds of fluorescence (see Material and methods). 'n' corresponds to the number of analyzed cells. Error bars are ± SEM; * $p \leq 0.05$; ** $p \leq 0.01$; *** $p \leq 0.001$.

significant (Fig 4). The exposure to $Mn^{2+}$ (Fig 5) modifies the size distribution of FA by decreasing the proportion of small adhesion sites while increasing that of large adhesion sites (Fig 5). Thus, CyaA triggers a drastic weakening in cell adhesion processes which can be partly reversible by exposure of cells to $Mn^{2+}$. In other words, manganese treatment partially counterbalances the changes in FA size and remodeling of F-actin structures induced by the toxin.

The effects of CyaA on the global shape of A549 cells were examined by analyzing images from different confocal planes. Statistics of cell spreading area and cell height for the different conditions of CyaA exposure tested reveals significant changes in cell shape after intoxication by the CyaA toxin (Fig 6A and 6B). The cell shape change can logically be related to the CyaA induced weakening process of adhesion presented in Figs 1 to 5 and suggests that CyaA triggers cell rounding. This is consistent with prior studies that showed a similar change in cell morphology, i.e., namely a rounding process, in various cultured cell lines—notably in type II alveolar cells [14].

To examine whether CyaA intoxication affects acto-myosin motors and notably intracellular cytoskeletal tension, we measured the changes in the cytoskeleton (CSK) stiffness. We analyzed the CSK stiffness by using the Magnetic Twisting Cytometry (MTC) technique. Ferromagnetic microbeads coated with an integrin specific ligand, (i.e., fibronectin), are used to probe the mechanical (rheological) properties of the cells when a torque is exerted by an externally applied magnetic field [27, 28].

CSK stiffness measurements performed after 30 min (Fig 7A) and 24 hrs (Fig 7B) of cell exposure to different toxin concentrations reveal a CyaA-dependent increase in CSK stiffness of up to 45% (as compared to that of control cells) after 30 min-exposure to the highest CyaA

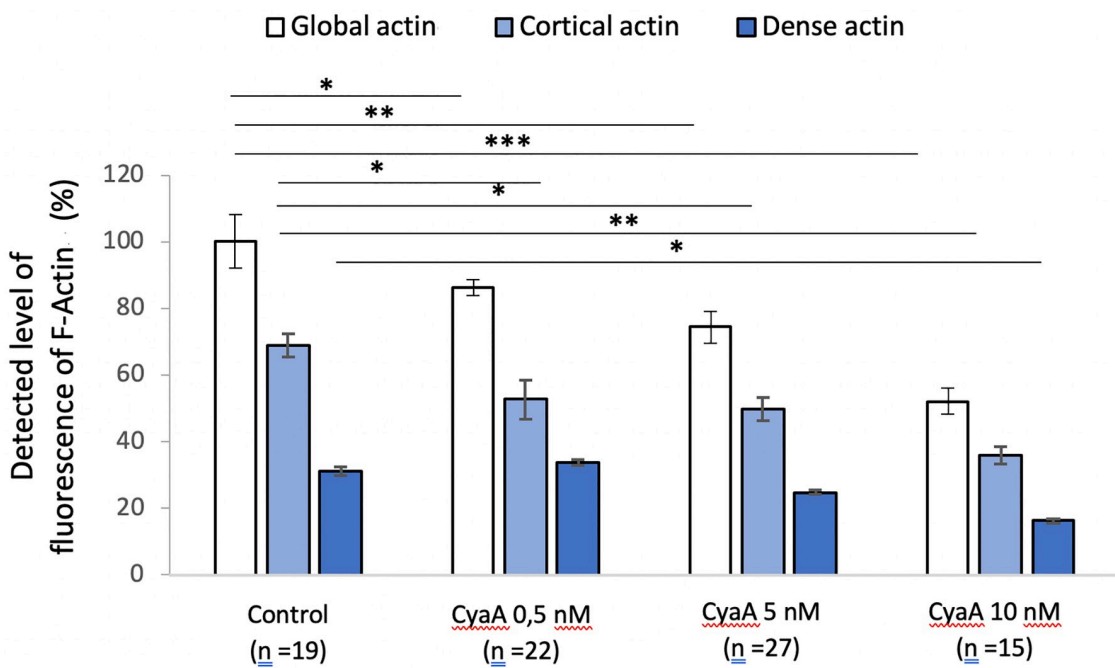

**Fig 3. Quantification of fluorescence levels of global, cortical and dense F-actin in A549 cells after 60 min of CyaA followed by 30 min of $Mn^{2+}$ exposure.** The bar graph provides a quantification of F-actin fluorescence in A549 cells before (control conditions) and after 60 min exposure to the indicated concentrations of CyaA, then, after washing, a 30 min exposure to $Mn^{2+}$ at 0.5mM. The specific distribution of the cortical (submembranous) and dense (stress fibers) actin structures were analyzed by using intermediate thresholds of fluorescence (see Material and methods). 'n' corresponds to the number of analyzed cells. Error bars are ± SEM; $^{*}$ $p \leq 0.05$; $^{**}$ $p \leq 0.01$; $^{***}$ $p \leq 0.001$.

concentration (10nM). This increase remains very significant even after 24 hrs-exposure to CyaA. Noteworthy, the significant decay in CSK stiffness observed after CytoD treatment confirms that the measured stiffness is actin-CSK specific as expected by the use of fibronectin-coated microbeads functionalized for integrins (see Material and methods). Note that changes in cell mechanical properties measured by MTC which are also known to reflect changes in cytoskeleton internal tension [34] are closely related to cell shape [37, 38].

The discrepancies between CytoD and CyaA effects reveal the high specificity of the CyaA-induced changes in cell morphology and mechanics. It is is likely due to differences in the biological processes induced by the chemical drug (CytoD) and the pathogenic toxin (CyaA): CytoD alters both the deep and the cortical components of the CSK but the deep component is particularly modified [39]. By contrast, CyaA exposure alters preferentially the cortical CSK component as shown above (Fig 2) and also by our prior AFM (Atomic Force Microscope) data showing an increase in the indented surface Young modulus [16].

Cell migration and tissue repair observed during a wound healing test is a hallmark of functional integrity for tissue cells since it involves fundamental cell processes such as adhesion, locomotion, cytoskeleton mechanical properties including intracellular tension, actin polymerization and actomyosin motors activation. We therefore explored whether CyaA intoxication might affect these processes. The wound healing test was carried out on monolayers of A549 alveolar epithelial cells previously exposed for one hour to different concentrations of CyaA (0.5, 5, or 10nM) and compared to control conditions (no toxin). The cell monolayers were then scratched with a pipette tip of 5 μm diameter to create a wound, and cell migration and repair were followed by microscopy until complete closure of the wounds (the repair area

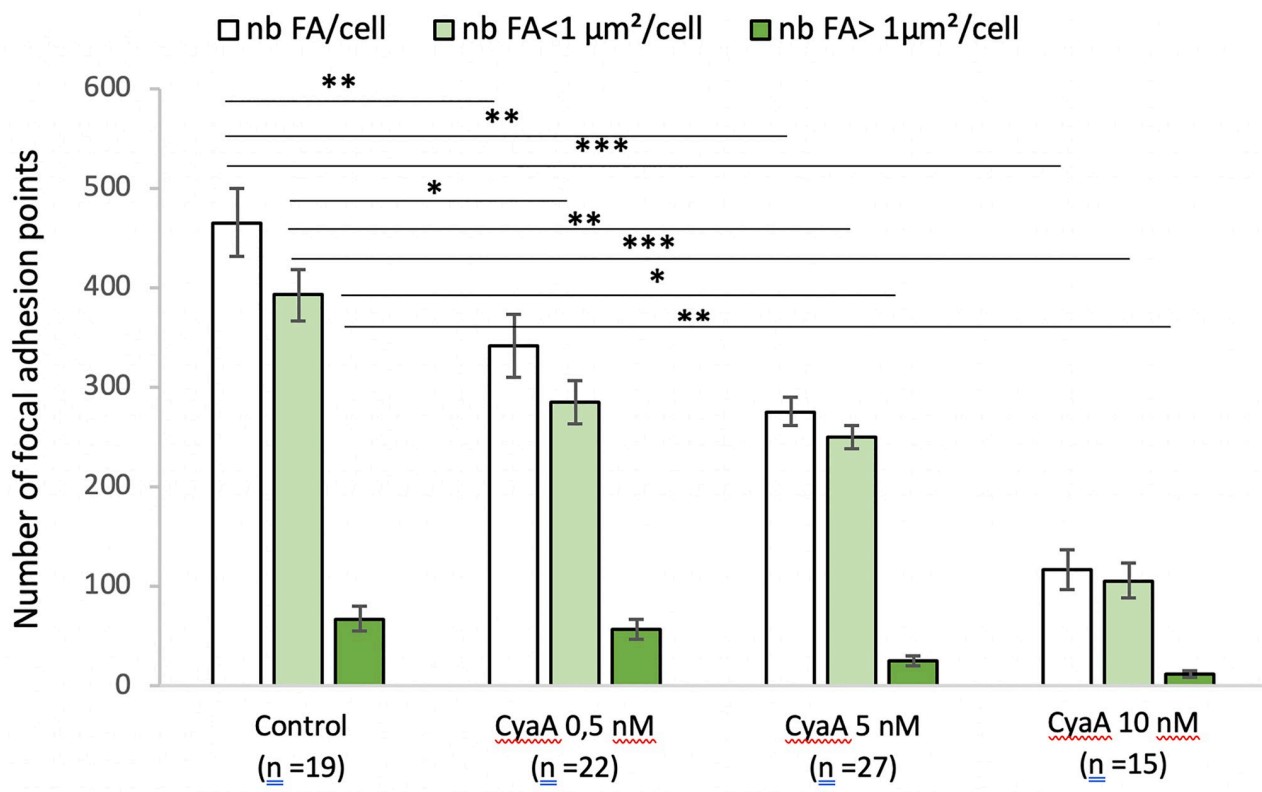

**Fig 4. Quantification of the number and size of focal adhesion points of A549 cells exposed to CyaA.** This bar graph shows the numbers of focal adhesion (FA) point per cell and their distribution in terms of size (surface areas of FA below and above 1 μm$^2$) in A549 cells incubated for 60 min with (or without control conditions) the indicated concentrations of CyaA. 'n' corresponds to the number of analyzed cells. Error bars are ± SEM; $^*$ $p \leq 0.05$; $^{**}$ $p \leq 0.01$; $^{***}$ $p \leq 0.001$.

were evaluated every 3 hrs up to 40 hrs). Results presented in Fig 8A and 8B reveal that CyaA intoxication drastically slows down both cell migration and overall wound repair. At the low toxin concentration of 0.5nM, the repair area is significantly reduced at all times tested. At the highest concentration tested (10nM), the repair is almost totally abolished (> 95%). However, the viability of A549 cells exposed to the highest CyaA concentrations is significantly altered upon prolonged incubation time (40 hrs) (S3 Fig) and this could explain in large part the dramatic decrease of wound repair observed in these conditions. As expected, the enzymatically inactive CyaA variant CyaAE5 had no effect on the cell migration nor wound repair (Fig 8C). Altogether, our present results indicate that the increase in intracellular cAMP elicited by the CyaA toxin directly impairs the cytoskeletal and mechanical properties of the intoxicated A549 alveolar epithelial cells and reduce their migration and wound healing capacities. As these effects are observed at the low toxin concentrations (0.5nM) that might be produced in vivo during *B. pertussis* infection [13], our results suggest that the CyaA could also contribute to the local alteration of the epithelial barrier of the respiratory tract, that is an hallmark of *pertussis*.

## Discussion and perspectives

While the infection by CyaA of myeloid cells such as macrophages, neutrophils, dendritic cells, natural killer cells, has been extensively studied [4, 40–42], much less is known about the intoxication of non-inflammatory cells lacking the CD11b/CD18 receptor. In these cells, CyaA

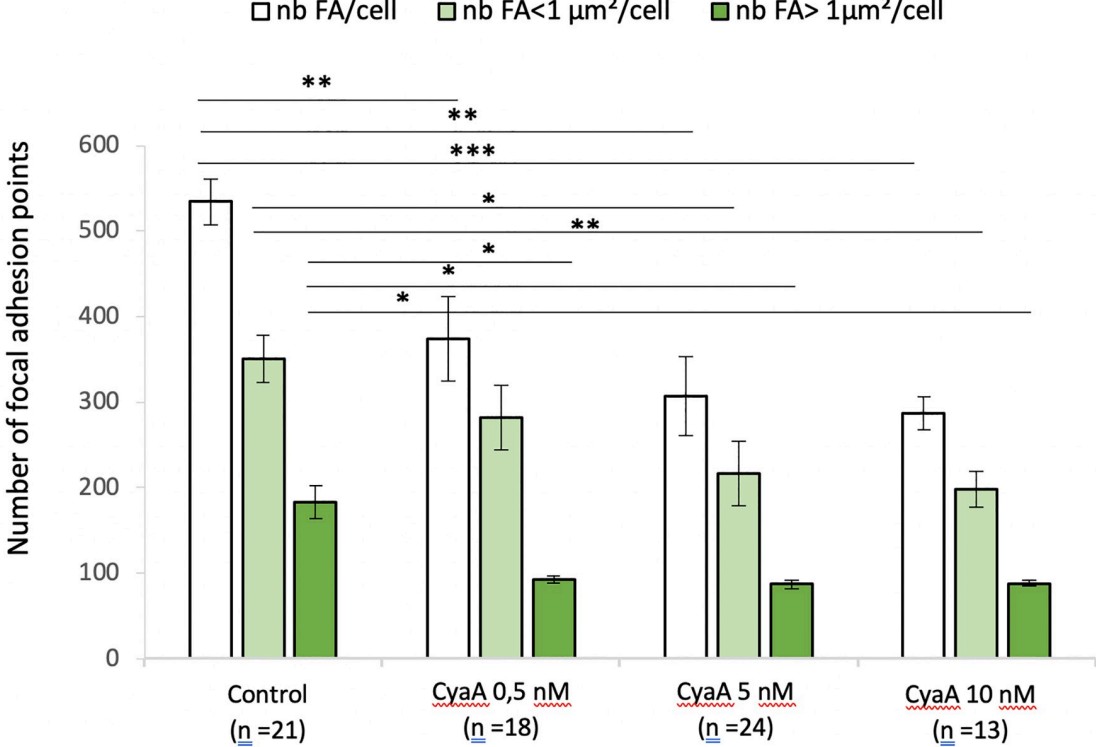

**Fig 5. Quantification of the number and size of focal adhesion points of A549 cells exposed to CyaA followed by 30 min of Mn²⁺ exposure.** This bar graph shows the numbers of focal adhesion (FA) point per cell and their distribution in terms of size (surface areas of FA below and above 1 μm²) in A549 cells before (control conditions) and after 60 min exposure to the indicated concentrations of CyaA, then, after washing, a 30 min exposure to $Mn^{2+}$ at 0.5mM. 'n' corresponds to the number of analyzed cells. Error bars are ± SEM; * $p \leq 0.05$; ** $p \leq 0.01$; *** $p \leq 0.001$.

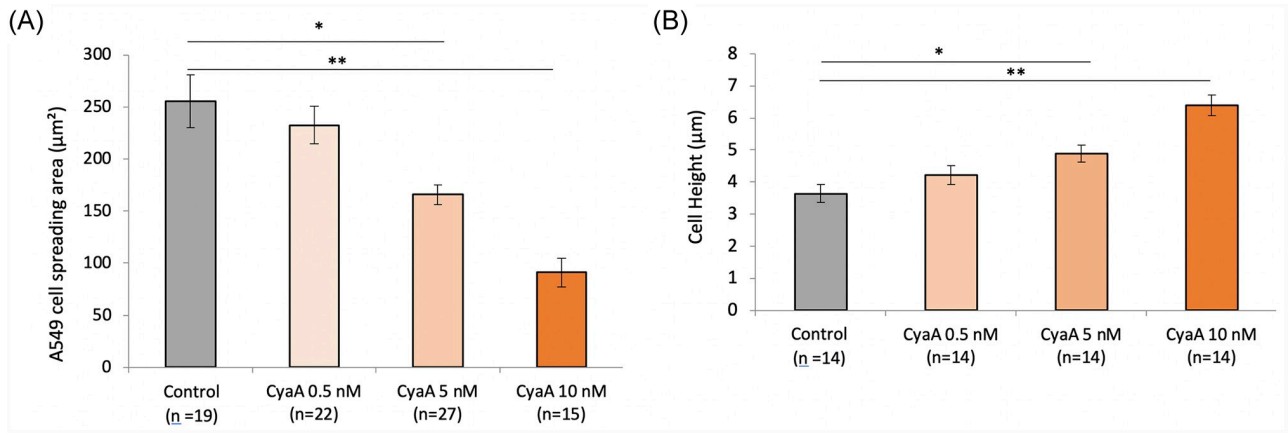

**Fig 6. Effect of CyaA on A549 cell rounding.** A549 cells were exposed for 60 min to the indicated concentrations of CyaA toxin (control = no CyaA) and analyzed by confocal microscopy to determine (A) the cell spreading area (in μm²) which is measured in the basal plane of the *z*-stack confocal images and (B) cell height (in μm) which is measured from the distance separating the basal and apical planes of the *z*-stack confocal images. 'n' corresponds to the number of analyzed cells. Differences between the different groups are quantified by ANOVA test compared to the control conditions. Error bars are ± SEM; * $p \leq 0.05$; ** $p \leq 0.01$; *** $p \leq 0.001$.

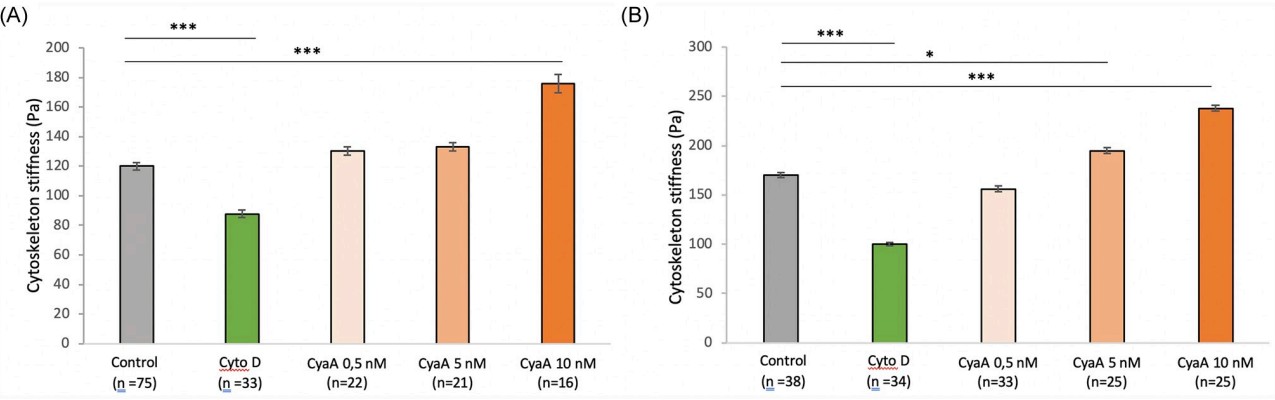

**Fig 7. Effect of CyaA on the cytoskeleton stiffness of A549 cells.** A549 cells were exposed for 60 min (A) or 24 hrs (B) to the indicated concentrations of CyaA toxin (control = no CyaA) or 10 μM of cytochalasin D (an actin-depolymerizing drug). The CSK stiffness is assessed using the viscoelastic solid-like model (simple Voigt) as described in 'Material and Methods'. Each MTC measurement is performed on a wide number of cells ($7 \times 10^4$) probed with an even higher number of beads (about twice). The 'n' corresponds to the number of wells. Differences in cytoskeleton stiffness observed between the different groups are quantified by ANOVA test compared to the control conditions. Error bars are ± SEM; * $p \leq 0.05$; ** $p \leq 0.01$; *** $p \leq 0.001$.

is able to directly interact with the plasma membrane and translocate its catalytic domain across the lipid bilayer although the precise molecular mechanism of intoxication remains elusive [1, 8, 43, 44].

We previously showed and we confirm here that CyaA can efficiently invade A549 alveolar epithelial cells to trigger a rapid and massive increase in intracellular cAMP. Our present data show that in these cells, CyaA induces a remodeling of actin cytoskeleton, a rapid decrease in size and number of FA sites, resulting in cell rounding and a significant increase in CSK stiffness. These modifications drastically impair the migration capabilities of A549 cells and their ability to repair wounds made in a monolayer of these epithelial cells. These alterations were observed at the lowest concentration (0.5nM) of CyaA. Importantly Eby *et al.* [13] reported that such concentrations might be reached locally at the bacterium-target cell interface during *B. pertussis* infection of the respiratory tract.

It has already been shown that CyaA invasion results in a deep remodeling of the actinic CSK localized essentially under the plasma membrane [9, 45]. Present data obtained on cellular adhesion as well as previous data obtained on early adhesion in the same cellular system [16] show that CyaA significantly weakens the adhesion system, suggesting that mechano-transduction signaling pathways are affected during the intoxication process.

In our previous study on the early development of adhesion sites in intoxicated cells [16], CyaA was reported to reinforce adhesion by decreasing the strength of newly formed integrin bonds as well as their mutual association through clustering. Thus, alteration of adhesion process and weakening of cell-matrix attachments can be considered as the hallmark of initial intoxication. Here, we show that CyaA intoxication is responsible for a cell rounding process that is likely resulting from the early alteration of adhesion. Such CyaA-induced morphological changes have already been documented in a variety of cultured cell lines [14]. When cells round and cystokeleton is deformed, filamin protein A (an actin-binding protein that cross-links F-actin) binds to p190RhoGAP and prevents its accumulation in lipid raft. p190RhoGAP is a RhoGAP protein which mediates integrin ligation-dependent inactivation of Rho [46]. Thus, in rounded cells, p190RhoGAP is inactive and hence the Rho activity is presumed to be high [47]. By contrast, in spread cells, p190RhoGAP accumulates in lipid rafts and become active leading to the suppression of Rho activity. Note that Rho's ability to increase myosin

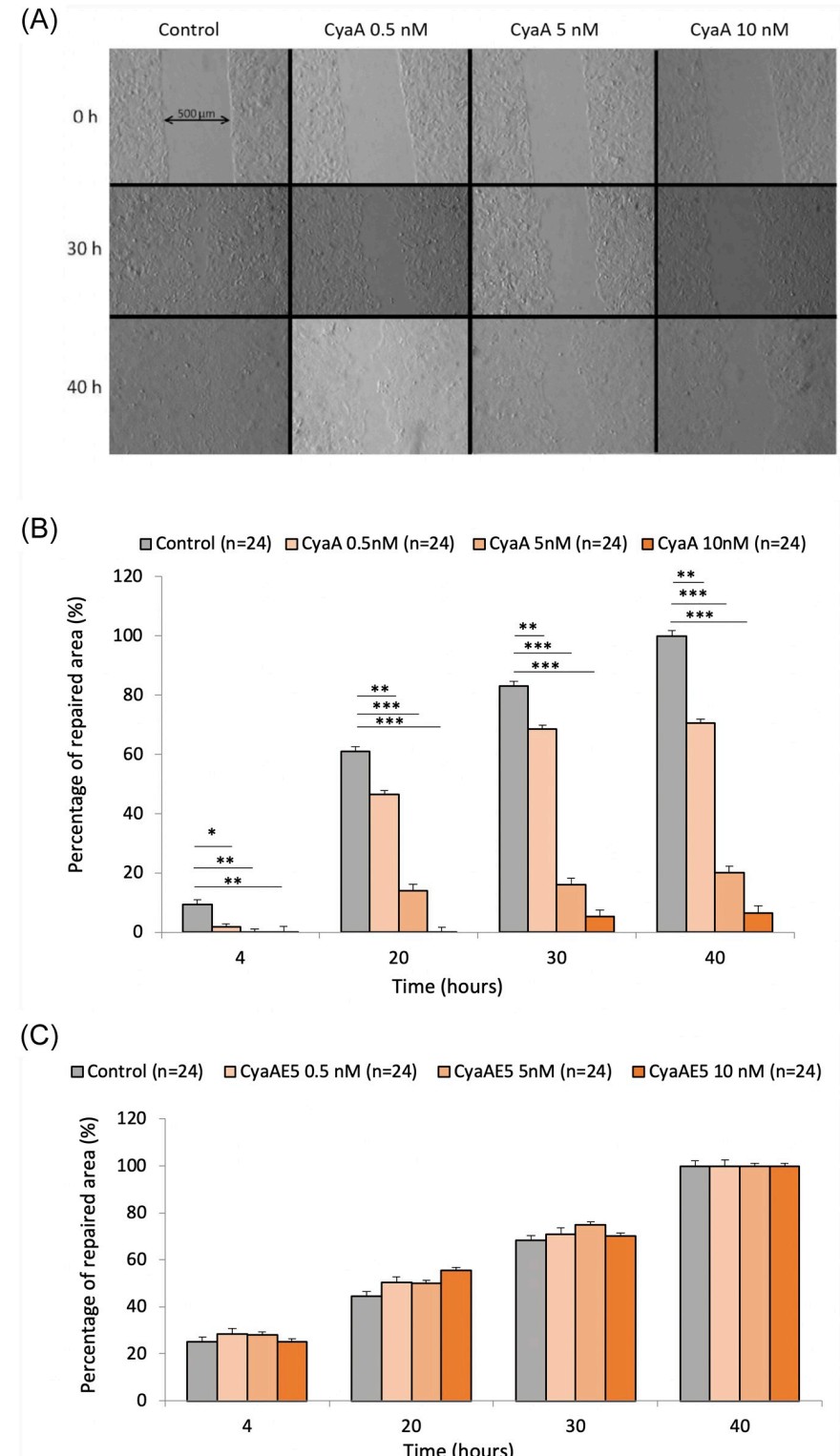

**Fig 8. Effect of different concentrations of the CyaA toxin on wound repair of the A549 cell monolayers.** Cells were grown to form a confluent monolayer and exposed during 1 hr to the indicated concentrations of CyaA toxin (control: no added CyaA). The cell monolayers were then scratched with a pipette tip (5 μm-distal diameter) to create a wound. After washing, cells were further grown in DMEM medium and the repair of the wounds was monitored by microscopy imaging (the wound area was measured with Image J) every 3 hrs until complete closure. Images of the

wound healing are shown in (A). They are obtained at 0 hr, 30 hrs, and 40 hrs, for control and for the 3 concentrations of CyaA tested (0.5, 5, 10nM). The initial wideness of the wound is about 500 m (see double arrow). The data in (B) correspond to the percentage of the repaired area observed at different times (4, 20, 30, 40 hrs) until the complete repair of the wound that occurs in control after 40 hrs (100% means fully repaired area). Control = no CyaA added. The data in (C) correspond to the wound repair of A549 cell monolayers exposed to the enzymatically inactive CyaAE5 protein instead of the wild-type CyaA toxin. Control = no added protein. 'n' corresponds to the number of analyzed wells. Error bars are ± SEM; * $p \leq 0.05$; ** $p \leq 0.01$; *** $p \leq 0.001$.

light chain phosphorylation through activation of Rho-associated kinase (ROCK) may feed-back to further activate Rho by increasing level of tension within the cytoskeleton [48, 49]. The increase in CSK stifffness presently observed in intoxicated cells is consistent with such an assumption. Activation of Rho GTPases in adhesion signaling involves extensive crosstalk between integrins, Src family kinases, and between individual Rho GTPases themselves [50, 51]. As a matter of fact, changes in cell shape are classically converted into changes in intracel-lular biochemistry leading to changes in actin-cytoskeleton-dependent control of Rho GTPase activity [52, 53]. Therefore, the balance between RhoA and/or Rac1 which is susceptible to affect cytoskeletal structure and cell mechanical properties [54, 55] is likely modified, primarily by the toxin induced change in cAMP levels [56] and secondarily by the CyaA-induced adhe-sion remodeling which affects cell shape (see adhesion data presently reported in Fig 4 and results obtained in previous studies [14]). By contrast, Kamanova *et al.* [9] reported that in macrophages the increase in cAMP elicited by CyaA causes transient RhoA inactivation that triggers actin CSK rearrangements and phagocytosis inhibition. Interestingly, Maddugoda *et al.* [57] also reported that another bacterial adenylate cyclase toxin the edema factor (EF) from *Bacillus anthracis* edema factor, as well as CyaA, could damage endothelium barriers by the formation of transendothelial cell macroaperture tunnels (TEMs) that directly result from cAMP-induced alterations of the actin cytoskeleton.

It is also recognized that migration, control of morphogenesis, and cell polarity are tightly controlled by Rho GTPases [58]. Present data show that these cellular functions are deeply affected by the toxin. The cytotoxic activity of the toxins caused by actin depolymerisation has already been explained by the role of certain Rho GTPases [59]; Maturation of integrin-medi-ated adhesion is controlled by the interplay between RhoA and Rac1 activations [46, 50]. Early adhesion, i.e., adhesion sites at the onset of maturation, is dependent on Rac1 activation and on a parallel inhibition of RhoA activity [60]. On the other hand, mature focal adhesion (FA) and associated tensed CSK elements require a high RhoA activity and Rac1 inhibition [61].

Overall, the toxin-induced cellular alterations experimentally observed in our A549 cells are consistent with the assumption of the maintenance of RhoA activity. RhoA is the upstream activator of the serine / threonine kinase RHOK (Rho-kinase) which in cooperation with scaf-fold protein mDia regulate actin-myosin assembly and contractility as well as actin polymeri-zation resulting in modifications in stress fiber formation and cell motility. We indeed observed on cellular images a persistence of stress fibers up to the highest CyaA concentration, while cytoskeleton stiffness was continuously increased as CyaA concentration was increased. On the other hand, cellular migration was significantly affected after CyaA intoxication in a concentration-dependent manner which suggests that the migratory phenotype which requires Rac1 activation might be limited by intoxication [60]. Moreover, it has been shown that cAMP modulates cell morphology by inhibiting a Rac-dependent signaling pathway [56]. Similarly, the results obtained in our previous study on the early development of adhesion while the same cells were intoxicated with the same toxin [16], have shown a reduction in chemical energy of individual integrin-RGD ligand bonds as well as a diminution by two of integrin association at the level of newly formed adhesion sites (clustering). For these two

processes, the role of Rac 1 is fundamental [62] and its potential defect in parallel to the activation of RhoA could be the common cause of these results. Yet, this presumed molecular mechanism remains to be evaluated in future studies.

Our present results are in good agreement with prior studies that showed that A549 cells treated with cAMP-increasing molecules (forskolin, isoproterenol, dibutyryl cAMP) display reduced cell migration and wound healing [63]. More precisely, agents that activate the cAMP/PKA/CREB pathway can lead to increased expression of Nm23-H1/2, a potential metastasis suppressor which has the ability to down-regulate metastasis formation independently of primary tumor size [64]. As a matter of fact, RGS19, a member of the regulators of G protein signaling (RGS proteins), appears to suppress tumorigenesis via upregulating Nm23-H1/2 [65, 66]. Activation of the protein kinase A (PKA) attenuated cancer cell migration in wound healing and transwell assays through a PKA-dependent mechanism for controlling Nm23-H1/2 expression [63]. Conversely, inhibition of protein kinase A (PKA) suppressed the phosphorylation of CREB and reduced the expression of Nm23-H1/2. Furthermore, in Chen et al. [67], cAMP inhibits the migration of mouse embryonic fibroblast (MEF) cells and highly invasive (4T1) mouse breast tumor cells by interfering with Rac-induced lamellipodium formation at the leading edge during cell migration.

Another important aspect that may explain the efficient effects of CyaA on the actin cytoskeleton remodeling and mechanical perturbations of the A549 cells is cAMP compartmentalization. Indeed, the specific pathway of CyaA entry, that involves direct translocation of the toxin catalytic domain across the plasma membrane of the intoxicated cells, results in a rapid and preferential production of cAMP in the vicinity of the plasma membrane with a delayed and lower cAMP signal in a perinuclear area [68]. It is well known that microdomains of cAMP have pronounced effects on various signaling processes and major physiological functions in a wide variety of cells [69]. For CyaA, this has been particularly well characterized in T cells where the toxin was shown to efficiently disrupt the immunological synapse and redistribute a number of essential players in the T-cell activation cascade [70, 71]. Remarkably, these effects are not elicited by another bacterial toxin, the anthrax edema factor that produces similar cytosolic levels of cAMP but originating from a perinuclear localization—where the toxin is released from late endosomes [71]. Hasan *et al.* recently reported similar findings on human monocytes [72]. It will be interesting to further delineate the precise role of cAMP compartmentalization in the biological effects of CyaA on the actin cytoskeleton remodeling and mechanical perturbations of epithelial cells evidenced here.

## Conclusion

In summary, we show here that the CyaA toxin is able to efficiently target epithelial cells where it triggers a rapid and large increase in intracellular cAMP. CyaA elicits various structural and functional modifications of the cells, including cell rounding, weakening of adhesions, remodeling of actin structures and of cytoskeleton stiffness. This leads to a drastic inhibition of cell migration and wound repair capabilities. These data support the hypothesis that CyaA, in addition to its critical role in blunting the innate immune responses, may also contribute, likely in synergistic manner with other virulence factors, to the local alterations of the epithelial barriers of the respiratory tract, a pathognomonic feature of *B. pertussis* infection.

## Supporting information

**S1 Fig. Viability assays of A549 cells exposed to CyaA toxin.** A549 cells were grown in monolayer to 90% of confluence and then incubated for 60 min with the indicated concentrations of CyaA. MTT was then added at 0.25 μg/ml and cells were further incubated 4 hrs at

37°C. The medium was removed and replaced by 200 μL of DMSO and the optical density at 550nm was recorded on a microplate reader. Control corresponds to cells incubated in similar conditions without CyaA. Error bars are ± SEM; * $p \leq 0.05$; ** $p \leq 0.01$; *** $p \leq 0.001$. These data show that the viability of A549 cells is not significantly affected when they are exposed during 1 hr to CyaA concentrations lower than 3nM, while it is drastically reduced at CyaA concentrations above 5nM.
(TIF)

**S2 Fig. Intracellular cAMP measurements in A549 cells exposed to either CyaA or CyaAE5 toxins.** Intracellular cAMP is measured by ELISA assay in A549 cells exposed to CyaA or to CyaAE5, a CyaA variant lacking enzymatic activity, at concentrations 0.5; 5 and 10nM and for 15, 30, and 60 min (n = 12 wells). Control conditions correspond to cells incubated without toxin. Error bars are ± SEM; * $p \leq 0.05$; ** $p \leq 0.01$; *** $p \leq 0.001$. These data show that even the lowest CyaA concentration (0.5nM) triggers a large increase in intracellular cAMP, that can be observed at the shortest exposure time (15 min) while very high cAMP levels can be reached observed at higher CyaA concentrations. As expected, no significant changes in intra-cellular cAMP levels are observed when cells are incubated with the enzymatically inactive toxin, CyaAE5.
(TIF)

**S3 Fig. Viability of A549 cells exposed to CyaA.** Viability assays performed by Trypan blue over 40 hrs on A549 cells in control conditions and after 1 hr of exposure time to different CyaA concentrations (0.5, 5 and 10 nM) (n = 3 wells). The test durations (4, 20, 30, 40 hrs) correspond to the times used for migration-repair experiments. The bar graph shows that the cell viability decreases with increasing CyaA concentration as well as with increasing test duration in many cases. * $p \leq 0.05$; ** $p \leq 0.01$; *** $p \leq 0.001$.
(TIF)

## Acknowledgments

The authors thank Emilie Bequignon, André Coste, Estelle Escudier, Jean-François Papon, Sofia Andre Dias, and Ngoc Minh Nguyen for their helpful advices and fruitful discussions.

## Author Contributions

**Conceptualization:** Daniel Ladant, Emmanuelle Planus, Alexandre Chenal, Daniel Isabey.

**Data curation:** Christelle Angely, Alexandre Chenal, Daniel Isabey.

**Formal analysis:** Christelle Angely, Daniel Ladant, Daniel Isabey.

**Funding acquisition:** Bruno Louis, Daniel Isabey.

**Investigation:** Christelle Angely, Daniel Ladant, Emmanuelle Planus, Alexandre Chenal, Daniel Isabey.

**Methodology:** Christelle Angely, Daniel Ladant, Emmanuelle Planus, Alexandre Chenal, Daniel Isabey.

**Project administration:** Bruno Louis, Daniel Isabey.

**Resources:** Bruno Louis, Daniel Isabey.

**Supervision:** Daniel Ladant, Emmanuelle Planus, Daniel Isabey.

**Validation:** Christelle Angely.

**Writing – original draft:** Christelle Angely, Daniel Ladant, Emmanuelle Planus, Marcel Filoche, Alexandre Chenal, Daniel Isabey.

**Writing – review & editing:** Christelle Angely, Daniel Ladant, Emmanuelle Planus, Marcel Filoche, Alexandre Chenal, Daniel Isabey.

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
