## [Decision Letter · Decision Letter 0]

12 Mar 2020

PONE-D-20-01560

Functional and structural consequences of epithelial cell invasion by Bordetella pertussis adenylate cyclase toxin

PLOS ONE

Dear Dr Daniel Isabey,

Thank you for submitting your manuscript to PLOS ONE. After careful consideration, we feel that it has merit but does not fully meet PLOS ONE’s publication criteria as it currently stands. Therefore, we invite you to submit a revised version of the manuscript that addresses the points raised during the review process.

We would appreciate receiving your revised manuscript by April 10. To enhance the reproducibility of your results, we recommend that if applicable you deposit your laboratory protocols in protocols.io, where a protocol can be assigned its own identifier (DOI) such that it can be cited independently in the future. For instructions see: http://journals.plos.org/plosone/s/submission-guidelines#loc-laboratory-protocols

We look forward to receiving your revised manuscript.

Kind regards,

Daniela Flavia Hozbor

Academic Editor

PLOS ONE

Journal Requirements:

https://www.tandfonline.com/doi/full/10.4161/sgtp.27958

https://www.sciencedirect.com/science/article/pii/B9780120884452500263?via%3Dihub

In your revision ensure you cite all your sources (including your own works), and quote or rephrase any duplicated text outside the methods section. Further consideration is dependent on these concerns being addressed.

Reviewers' comments:

Reviewer's Responses to Questions

**Comments to the Author**

1. Is the manuscript technically sound, and do the data support the conclusions?

Reviewer #1: Partly

Reviewer #2: Yes

Reviewer #3: Yes

2. Has the statistical analysis been performed appropriately and rigorously? 

Reviewer #1: Yes

Reviewer #2: Yes

Reviewer #3: Yes

3. Have the authors made all data underlying the findings in their manuscript fully available?

Reviewer #1: Yes

Reviewer #2: Yes

Reviewer #3: Yes

4. Is the manuscript presented in an intelligible fashion and written in standard English?

Reviewer #1: Yes

Reviewer #2: Yes

Reviewer #3: Yes

5. Review Comments to the Author

Reviewer #1: In the manuscript “Functional and structural consequences of epithelial cell invasion by Bordetella pertussis adenylate cyclase toxin” the authors describe major effects of CyaA on epithelial cell structure and function. This work builds incrementally on earlier data to show that these cells, when treated with CyaA, also display altered cell morphology, cytoskeletal stiffness and impaired wound healing capacity. The manuscript is clearly written, logically presented and nicely demonstrates the dramatic effect of CyaA intoxication on epithelial cells, a cell type not typically associated with CyaA-mediated pathologies. However, the authors should be careful not to attribute CyaA-mediated effects solely to elevated cAMP in those studies in which a control, demonstrating the cAMP-dependence, is not included. Major and minor comments provided below.

Major comments:

1. Experiments performed in Fig 7 could be improved with the addition of cytochalasin D to the CyaA treated cells to demonstrate the actin dependence of the CyaA-mediated effects

2. In earlier work, this group demonstrated that CyaA reduced the viability of A549 cells within 15 min of exposure to 10 nM CyaA, while a non-significant reduction was observed at 60 min with 0.5 nM CyaA. And, in Fig S1 CyaA is again demonstrated to reduce cell viability. The results presented in Fig 8 represent cells treated for 60 min with CyaA and measurements are displayed from 4 hours. In order to be consistent with the wound repair parameters, it would be good to include the cell viability as assessed at similar time points to those used in Fig 8. There is presumably significant loss in cell viability in treated cells in this assay. The authors should consider that a reduction in wound healing may not be solely attributable to a disrupted actin CSK. Loss in cell viability should be mentioned as a possible alternative mechanism or loss in cell viability should be controlled for in the experimental design.

3. In the conclusions, it is stated that “the large increase in intracellular cAMP elicits various structural and functional modifications of the cells, including cell rounding, weakening of adhesions, remodeling of actin structures and of cytoskeleton stiffness” and that this “leads to a drastic inhibition of cell migration and wound repair capabilities”. However, in this manuscript, only the actions of CyaA on wound repair were proven to be cAMP-dependent, as the impact of CyaA was ablated when an enzymatically inactive CyaAE5 was added in place of active CyaA. In order to attribute the actions of CyaA to elevated cAMP levels for the remaining parameters, the authors must repeat these studies using CyaAE5 or with treatment targeting the reduction of cAMP e.g. Rp-8-Br-cAMPs

Minor comments:

1. In the materials and methods, the description of A549 culture is extensive and may be reduced

2. Line 80, spell out the LQ amino acids when first described

3. Line 103, provide first description for “ACD”

4. Line 108, provide first description for “SVF”

5. Provide abbreviation for CSK at first usage (line 131 not line 240)

6. Line 156, provide information on how cells were selected for structural analysis e.g. 5 random fields etc.

7. Section “Measure of CSK stiffness by MTC” identify the instrument used for this method

8. Figure 8 could be improved by the addition of images of the end point of the wound healing assay; demonstrating a healed zone for control cells vs minimal healing by the CyaA-treated cells.

9. In support of a cAMP-mediated role in impaired wound repair, it may be worth mentioning that A549 cells treated with cAMP-increasing molecules (forskolin, isoproterenol, dibutyryl cAMP) display reduced cell migration and wound healing (See RGS19 upregulates Nm23-H1/2 metastasis suppressors by transcriptional activation via the cAMP/PKA/CREB pathway)

10. A549 cells possess some phenotypic plasticity, it could be interesting to determine whether high cAMP levels alter the cell differentiation state. For example, if it drives the cell away from a proliferative and reparative alveolar type 2 epithelial phenotype and towards a more quiescent AEC type 1 or perhaps even a neuroendocrine-like cell (high cAMP levels promote neuroendocrine differentiation in A549 cells, reference: Effect of A549 neuroendocrine differentiation on cytotoxic immune response)

Reviewer #2: Manuscript by Angely et al. describes structural and functional effects of Bordetella adenylate cyclase toxin on cytoskeleton of A549 alveolar epithelial cells. The main message of manuscript is far from being novel, nevertheless the manuscript is technically sound, and seems to merit PlosOne publication criteria.

The only major concern is the cytotoxicity of 5nM and 10 nM CyaA (Suppl.Fig.1) and the use of these CyaA concentrations in the other assays in addition to 0.5nM. One may then ask wheter there is actually an effect of CyaA on cytoskeleton stiffness (Fig7) or whether the effect on cell migration is so dramatic (Fig8).

Minor comment: Authors may want to explain better the rationele behind using manganese in their experiments.

Reviewer #3: The authors present a data concerning the effects of CyaA (ACT) toxin on epithelial cells. CyaA has been extensively characterized in regards to its effects on phagocytes and other immune cells. The manuscript shows data speaking to the structural effects of high cAMP on cells in vitro. Multiple assays are used to backup the findings. Overall the manuscript it easy to read and clear.

Major concerns

- In the abstract one sentence reads: "We also show that, at the low concentrations that may be found in vivo during B. pertussis infection, CyaA impairs the migration and wound healing capacities of the intoxicated alveolar epithelial cells." This re viewer does not believe that data are fully available to determine the exact amount of CyaA in vivo during infection. Some studies have shown staining with anti-CyaA. Overall it is not clear how much CyaA is in the respiratory tract of experimental animals or infected humans. The authors should refrain for over-extrapolation of the data and revise the statement in the abstract to read "We also show that, at the low concentrations (insert amounts)...." It is appropriate for the authors to discuss CyaA in vivo concentrations in the discussion.

Minor concerns

none

6. PLOS authors have the option to publish the peer review history of their article (what does this mean?). If published, this will include your full peer review and any attached files.

Reviewer #1: Yes: Karen M Scanlon

Reviewer #2: No

Reviewer #3: No

---

## [Author Response · Author response to Decision Letter 0]

2 Apr 2020

Regarding the minor occurrences of overlapping text with the following previous publication(s):

1. https://www.tandfonline.com/doi/full/10.4161/sgtp.27958

2. https://www.sciencedirect.com/science/article/pii/B9780120884452500263?via%3Dihub

We corrected the relevant sentences for the first reference, rephrased them, and quoted the reference appropriately (line 361 of the manuscript). We were however not able to find any overlap with the second reference. We did not refer to nor used this reference to conduct our work.

---

## [Decision Letter · Decision Letter 1]

20 Apr 2020

Functional and structural consequences of epithelial cell invasion by Bordetella pertussis adenylate cyclase toxin

PONE-D-20-01560R1

Dear Dr. Daniel Isabey,

We are pleased to inform you that your manuscript has been judged scientifically suitable for publication and will be formally accepted for publication once it complies with all outstanding technical requirements.

With kind regards,

Daniela Flavia Hozbor

Academic Editor

PLOS ONE

Additional Editor Comments (optional):

Reviewers' comments:

Reviewer's Responses to Questions

**Comments to the Author**

1. If the authors have adequately addressed your comments raised in a previous round of review and you feel that this manuscript is now acceptable for publication, you may indicate that here to bypass the “Comments to the Author” section, enter your conflict of interest statement in the “Confidential to Editor” section, and submit your "Accept" recommendation.

Reviewer #1: All comments have been addressed

2. Is the manuscript technically sound, and do the data support the conclusions?

Reviewer #1: (No Response)

3. Has the statistical analysis been performed appropriately and rigorously? 

Reviewer #1: (No Response)

4. Have the authors made all data underlying the findings in their manuscript fully available?

Reviewer #1: (No Response)

5. Is the manuscript presented in an intelligible fashion and written in standard English?

Reviewer #1: (No Response)

6. Review Comments to the Author

Reviewer #1: (No Response)

7. PLOS authors have the option to publish the peer review history of their article (what does this mean?). If published, this will include your full peer review and any attached files.

Reviewer #1: Yes: Karen M Scanlon

---

## [Editor Report · Acceptance letter]

24 Apr 2020

PONE-D-20-01560R1 

Functional and structural consequences of epithelial cell invasion by *Bordetella pertussis* adenylate cyclase toxin 

Dear Dr. Isabey:

I am pleased to inform you that your manuscript has been deemed suitable for publication in PLOS ONE. Congratulations! Your manuscript is now with our production department. 

With kind regards,

on behalf of

Dr. Daniela Flavia Hozbor 

Academic Editor

PLOS ONE